# Impact of Intellectualization of a Zoo through a FCEM-AHP and IPA Approach

**Yuxuan Lin**  **and Ryosuke Shimoda \***

Graduate School of Horticulture, Chiba University, 648 Matsudo, Matsudo-City 271-8510, Japan
\* Correspondence: r.shimoda@chiba-u.jp

**Abstract:** As urbanization is growing faster, the term "Smart" is becoming more widely used. However, it is difficult to define how to objectively evaluate intellectualization. This study aims to explore an objective method of evaluating intellectualization in Japanese zoos and suggest project directions for their future development. First, we will define the unique Japanese zoo smart projects. Then, an analytic hierarchy process (AHP) will be used to determine the weights of each item, and the fuzzy comprehensive evaluation method (FCEM) will be used to evaluate the degree of construction. Finally, the strengths and weaknesses of each project will be analyzed by importance–performance analysis (IPA). The findings showed that the research subject, Ueno Zoo, is still in the early stage of smartening, and most of the items are not sufficient for users to have a full tourist experience. There is a need to increase the level of intellectualization and ease-of-use for the construction of the zoos of Tokyo. This study provides an objective approach for evaluating the intellectualization of zoos in Japan and provides a method of construction advice for intellectualization construction.

**Keywords:** smart zoo and smart city; intellectualization; fuzzy comprehensive evaluation method (FCEM); importance–performance analysis (IPA); smart zoo system of Japan (SZSOJ); Ueno Zoological Gardens



## 1. Introduction

### 1.1. Background

With the rapid development of cities and advances in technology, the construction of saturated metropolises requires paying increasing attention to the green-space environment and the life of citizens. The concept of the "Smart City" evolved from the concept of "Smart Earth", which was introduced by IBM in a theme report released in New York in 2008 [1]. According to the definition, a smart city is a technologically modern urban area that uses different types of electronic methods and sensors to collect specific data. The information gained from that data is used to manage assets, resources, and services efficiently; in return, such data is used to improve operations across the city. Smart cities use Information and Communication Technology (ICT) and Internet of Things (IoT) technologies to bring citizens a more informed and digital urban life. In this context, "Smart" has become one of the leading keywords for urban development, and the construction of intellectualization (in Japanese, 'sumāto-ka') is being used by various industries [2]. ICT, IoT, Artificial Intelligence (AI), 5G, and other keywords are entering the public's mind. Smart cities are increasingly in demand for platforms that use the IoT as an information network to deliver information constituting a big data substrate. Moreover, with growing/expanding urban development, a large number of "Smart Park" constructions are being conducted— for example, Haidian Park in Beijing [3], China, Xiangmi Park in Shenzhen [4], China, Arashiyama Park (Nakanoshima Area) in Kyoto [5], Japan, and Palace Site Historical Park in Nara [6], Japan. While smart parks provide better urban green-space life for citizens, the present study focuses on one particular type of park in this context: zoos. More targeted sources of visitor traffic and a richer ecological environment can make the intellectualization of zoos more meaningful.

Especially under the impact of the COVID-19 pandemic in recent years, zoos have struggled to operate. The impact is not only in the reduced number of visitors but also the reduced food for animals and lack of labor costs due to the declining birthrate and aging population, all of which affect the economic situation of a zoo. For example, feed fees at Tobu Zoological Park have increased by ~5–6% in 2020. In addition, Asahiyama Zoo was temporarily closed from April 2020 until January 2021. Since then, admission revenue has decreased by about 70% due to a sharp decline in inbound visitors from overseas.

To address these, a series of zoo-smartening projects have been tried in order to change the situation. For example, the company KDDI in Japan has developed the "one zoo" online zoo platform including famous zoos such as Asahiyama Zoo in Hokkaido and Tennoji Zoo in Osaka, where users can observe the animals online in real-time or donate funds to the animal protection association in the form of membership purchases, with the platform providing zoo tickets or souvenirs. Unfortunately, as of 31 May 2022, the project has been discontinued due to a lack of online activity. We tend to believe that the reason for this is that, although the developers have taken steps to smarten up the zoo tour experience, they have not considered the feedback from the user community on each of the smartening projects in a timely manner and lack an objective analysis.

We therefore focused our questions on obtaining timely feedback on the smart-zoo building process and exploring a stable way to understand the current level of zoo intellectualization and improvement measures. The study will help zoo managers to find suitable management methods and directions for improvement. Considering that smart zoos in Japan are at an early stage of development, timely feedback can help to find a more objective and full-character approach to intellectualization development in order to avoid a uniform and unadopted outcome. Therefore, this study will focus on the intellectualization process of smart zoos existing in society and discuss the phase feedback of current construction as well as ways for it to be improved.

The present study will investigate the existing smart devices in the target zoo for the degree of awareness of visitors, and will analyze the degree of its smart construction and the future construction investment required. The intellectualization of zoos will be quantitatively studied. Data will be collected and vectorized through a questionnaire based on two dimensions. The result of the vector calculation using the "FCEM (fuzzy comprehensive evaluation method)—AHP (analytical hierarchy) process analysis method" can determine the current intellectualization level of the target zoo, and the results of importance–performance analysis (IPA) will guide the future development direction. Finally, this study aims to make smart functions more widely available in zoos, and the evaluation standards of smart zoos will be unified to create better zoos.

*1.2. Literature Review*

At present, there is not wide discussion within the academic community regarding the concept of smart zoos; rather, the term "Intellectualization of zoos" (in Japanese, Dōbutsuen no sumāto-ka) has been used in the context of Smart Parks in Japan.

The "SMART PARK: A TOOLKIT" released by UCLA [7] provides a comprehensive explanation of smart parks. The paper establishes a model for evaluating smart parks based on spatial characteristics from the perspective of designers, park managers, advocates, and even those who simply wish to learn how technology can be incorporated into parks. Although the research model satisfactorily explains the specific program definitions, it lacks an objective data evaluation system. A systematic model for smart parks based on functions was also established in the "Research on the Construction Framework of Smart Park: A Case Study of Intelligent Renovation of Beijing Haidian Park" [3]. However, the study also lacks a survey of tourists' feelings and objective data. The article "How smart is your tourist attraction? Measuring tourist preferences of smart tourism attractions via an FCEM-AHP and IPA approach" [8] pioneered the use of FCEM-AHP and IPA in the field of smart parks to analyze the weighting of parks and tourism preferences. The study used a questionnaire to collect data, used AHP to determine the weight set, and finally

used a fuzzy comprehensive evaluation to derive the tourism preferences, strengths, and weaknesses of the whole park. The model of the study is very comprehensive, but there are also problems in the questionnaire. For example, the lack of project descriptions and illustrations results in interviewees having a limited understanding of the items as defined in the questionnaire. In addition, many projects in the study should be defined again because of changes over the past few years.

## 2. Materials and Methods

### 2.1. Study Area

This study is a quantitative study of zoos in Japan, which are in the process of intellectualization. Japan's zoos have been well built and operated for a certain number of years, and the Japanese people have a good understanding of zoos daily, which provides a good foundation for the study of smart zoos.

In this context, the research sites will be selected from among the zoo members of the Tokyo Zoological Park Society that are engaged in the process of intellectualization and building platforms. The Tokyo Zoological Park Society (Tokyo Dōbutsuen Kyōkai) is a non-profit organization that manages and operates the zoos and aquariums owned by the Tokyo Metropolitan Government: Ueno Zoological Gardens, Tama Zoological Park, Tokyo Sea Life Park, and Inokashira Park Zoo. In this study, the Ueno Zoological Gardens in the Ueno area, Tokyo—which is a member of the Tokyo Zoological Park Society—was defined as the research target. Ueno Zoological Gardens is the oldest zoo in Japan. Founded in 1882, it has grown over the years, expanding its area to 14.4 ha (35.6 acres), and has become the flagship zoo of the Japanese zoo world. Now, it is home to over 3000 animals from 400 different species and provides visitors with a learning experience about the diversity of animals as well as fun and enjoyment. The reasons for the selection are as follows.

1.  The zoo is located in the Ueno area in the center of the city of Tokyo, which has a greater flow of guests than the other three member locations.
2.  Longer construction time, historical and cultural influence, higher visibility, and better awareness among tourists and citizens.
3.  The Ueno area has more foreign visitors, which meets the future needs of the zoo for visitors and is more in line with the identity of the smart zoo.

### 2.2. Identifying Evaluation Items of Smart Zoo System of Japan (SZSOJ)

It can be argued that the concept of intellectualization has been introduced in Japanese zoos for only a short time, and is unique to Japanese urban life habits. Thus, this project will be defined based on the zoo's developed projects. Finally, 5 primary classification items and a total of 24 secondary classification items were identified. The conceptual definition of these items is summarized in Table 1.

**Table 1.** The original evaluation items of *SZSOJ* and references.

| No. | Factors | Factor Description | Sources |
|---|---|---|---|
| 1 | Official website function | Tourists are increasingly utilizing digital sources such as tourism websites in their information search and communication exchange. Their travel intentions are strengthened by the website features of a tourism destination. | Heung, 2003 [9]; Kaplanidou and Vogt, 2007 [10] |
| 2 | Online information access | More travelers are using the internet as a medium to search for tourism information and planning their trips. Travel information is now among the most popular and frequently visited types of information on the internet. | Law and Leung, 2000 [11]; Buhalis and Law, 2008 [12] |
| 3 | Event notice | Using mobile application(s), tourist attractions can broadcast event information to tourists. | Wang and Zhu, 2013 [13] |
| 4 | Message notification (e-mail from Tokyo Zoonet) | Zoo Express is an e-mail magazine providing the latest information and animal-related news from the Tokyo Metropolitan Zoo and Aquarium. | https://www.tokyo-zoo.net/express/index.html (accessed on 12 January 2023) |
| 5 | Electronic claim system | Tourism organizations should have an e-complaint handling system to provide channels for tourists' feedback and complaints. | Buhalis and Law, 2008 [12] |

**Table 1.** *Cont.*

| No. | Factors | Factor Description | Sources |
|---|---|---|---|
| 6 | Electronic ticket system | A mature scenic spot information system should have an electronic ticketing system that utilizes radio-frequency identification technology. | Guo, Liu, and Chai, 2014 [14] |
| 7 | Online booking | Using technologies such as Wi-Fi, global navigation satellite systems, geographic information systems, and global positioning systems to meet tourists' mobile reservation demands. | Wang and Wang, 2010 [15]; Buhalis and Law, 2008 [12] |
| 8 | Tourism blog | A recent survey found that tourists trusted more websites with reviews than professional guides and travel agencies, and that, far from being an irrelevant, blogs are often perceived to be more credible and trustworthy than traditional marketing communications. | Akehurst, 2009 [16] |
| 9 | Animal introduction platform | The animal introduction platform is full of information about zoos, aquariums, and living creatures. Users can make great use of the entire website for school education. | https://www.tokyo-zoo.net/zoovie/ueno (accessed on 12 January 2023) |
| 10 | VR sightseeing | A virtual travel community makes it easier for people to obtain information, maintain connections, develop relationships, and eventually make travel-related decisions through the mediation of computer bulletin boards and networks. | Stepchenkova, Mills, and Jiang, 2007 [17] |
| 11 | 3D model | The 3D model is a part of a digital twin which is a virtual representation of a real-world physical system or product (a physical twin), serving as its indistinguishable digital counterpart for practical purposes such as system simulation, integration, testing, monitoring, and maintenance. | https://my.matterport.com/show/?m=6B29NY93fGS (accessed on 12 January 2023) |
| 12 | Application | Mobile application(s) can improve the quality of customer service in theme or amusement parks by showing the exact location of rides and allowing users to check on waiting times at different attractions and make reservations. | Hafele, 2011 as cited in Liu and Law (2013) [18] |
| 13 | Digital map | Using global positioning systems, electronic maps and compasses can collect information from satellites and provide tourists with exact geographic positions and directions. | Hawkins, Leventhal, and Oden, 1996 [19] |
| 14 | Sightseeing route recommendation | People can design a tour plan with the collaborative tour-planner system in a collaborative manner, and design personal itineraries according to their own points of interest. | Kurata and Hara, 2013 [20] |
| 15 | Stamps | Collect stamps while visiting the animals around the zoo. Those who successfully collect all 10 stamps will receive an original animal image of the Ueno Zoo. | https://www.tokyo-zoo.net/zoo/ueno/TPnavi/index.html (accessed on 12 January 2023) |
| 16 | Information from QR code | Quick response codes can access information about nearby points of interest through mobile devices. | GSMA, 2012 [21] |
| 17 | Animal information service | Guests can quickly access the animal information they are looking for in multiple ways. On the phone, they can look up information about the ecology not only presented in text, but also with audio and video, and use it like a guide terminal. | https://www.tokyo-zoo.net/zoo/ueno/TPnavi/index.html (accessed on 12 January 2023) |
| 18 | Smart guide system | Guiding information services combine mobile geographic information systems and global positioning system techniques with location-based services to provide tourists with a better trip experience and a deeper understanding of the importance of the valuable landscape. | Chu, Lin, Chang, and Chen, 2011 [22] |
| 19 | Tourist spots recommendations | E-tourism-recommendation technologies can provide valuable information to tourists and help them discover and select the points of interest that best fit their preferences. | Buhalis and Law, 2008 [12] |
| 20 | Free Wi-Fi | Free Wi-Fi allows users to connect mobile devices such as personal digital assistants and mobile phones to the internet through wireless radio connection. It is now extensively used in hotels, airports, and cafes. | Buhalis and Law, 2008 [12] |
| 21 | Digital payment | Purchasing tourism products through mobile websites and application(s) is an emerging market. More and more tourists use mobile devices to plan, purchase, and enhance their travel experiences. | Huang, Li, and Li, 2013 [23] |
| 22 | Fundraising | A facility that enables visitors to make financial contributions to a foundation association operated by the Tokyo Zoological Park Society through the use of QR codes. | https://www.tokyo-zoo.net/fund/index.html (accessed on 12 January 2023) |
| 23 | Electronic information screen | Electronic touch screen technology can provide services to meet tourists' information needs 24 h a day. | Connell and Reynolds, 1999 [24] |
| 24 | Animal state observation (Monitoring) | The animals' living conditions are recorded in real-time using surveillance cameras and displayed through a display for visitors. | https://www.ueno-panda.jp/movie/ (accessed on 12 January 2023) |



*2.3. Data Collection*

The data was collected in the form of a Google questionnaire, and 101 valid responses were collected. Since answering the questionnaire requires the respondent's account to be logged in beforehand, this ensures the validity and authenticity of the questionnaire. The respondents were all graduate students in landscape architecture from Chiba University and the University of Tokyo, and were confirmed to have had a full tourist experience at Ueno Zoo before partaking in the questionnaire. Due to their academic qualifications, all participants could evaluate the smart zoo experience from a researcher's perspective, and the complete tourist experience ensured the validity of the questionnaire. The questionnaire is categorized in two levels of indicators (Level 1 and 2; see File S1 for questionnaire items) and items in both levels are assessed for importance and performance on a scale of 1–5 (1 = not at all important, 5 = very important in the importance assessment, and 1 = very poor, 5 = very good in the performance assessment). Each item was accompanied by a graphical description so that respondents would not misidentify the item. In addition, the questionnaire was tested for reliability and the results were used to ensure the quality of the questionnaire.

The data collected was applied in two parts. The first part was based on the importance ranking of the survey results as an objective data reference, and then we used the AHP to confirm the weight of each item, and finally applied these AHP-derived weights to the FCEM to obtain the results of the zoo's current construction effectiveness. The second part retained the original 1–5 rating data for FCEM, and used IPA testing to determine the current status of the overall intellectualization construction degree and each item in the specific zoo to advise on future zoo development.

*2.4. AHP*

The analytic hierarchy process (AHP), formally introduced in the mid-1970s by American operations researcher Thomas L. Saaty, is a systematic and hierarchical method of analysis that combines qualitative and quantitative approaches, and is a representative method for quantifying group decisions and weights [25]. The current study required an assessment of the relative weights of the items, so an AHP with a pair-wise comparison of the weights of each item was used for the analysis.

The relative importance of different criteria between each item can be derived from the importance ranking of the questionnaire, and since the process of pair-wise comparison requires importance comparison on a scale of 1–9, the items with importance ranking collected from the questionnaire are transformed into percentages with the scale of 1–9 to derive the importance judgment results of pair-wise comparison among all items. The scales of relative importance are shown in Table 2.

**Table 2.** Scales of relative importance.

| Scales of Relative Importance | Meaning |
| :---: | :---: |
| 1 | Equally important |
| 3 | Slightly important |
| 5 | Quite important |
| 7 | Obviously important |
| 9 | Absolutely important |
| 2, 4, 6, 8 | Intermediate scales |
| 1/3 | Slightly unimportant |
| 1/5 | Quite unimportant |
| 1/7 | Obviously unimportant |
| 1/9 | Absolutely unimportant |
| 1/2, 1/4, 1/6, 1/8 | Intermediate scales |

Each evaluated item set is also defined by U.
The classification is defined as follows:

$U = \{U_1, U_2, U_3, U_4, U_5\}$
$U_1 = \{U_{11}, U_{12}, U_{13}, U_{14}, U_{15}, U_{16}, U_{17}, U_{18}\}$
$U_2 = \{U_{21}, U_{22}, U_{23}\}$
$U_3 = \{U_{31}, U_{32}, U_{33}, U_{34}, U_{35}, U_{36}\}$
$U_4 = \{U_{41}, U_{42}\}$
$U_5 = \{U_{51}, U_{52}, U_{53}, U_{54}, U_{55}\}$

The defined items are analyzed by AHP according to the importance ranking and a judgment matrix is constructed. The maximum eigenvalue of the judgment matrix is calculated, and the eigenvector is the evaluation weight vector A. Finally, the objectivity and rationality of the judgment are ensured by the consistency test.

In the whole calculation, CI is the deviation consistency index of the judgment matrix, $CI = \frac{(\lambda - n)}{(n-1)}$; the larger the CI is, the worse the consistency of the judgment matrix, and when CI is 0, the judgment matrix has full consistency. CR is the consistency ratio, with a formula of $CR = \frac{CI}{RI}$, where RI is the average random consistency index. When CR < 0.1, the consistency of the judgment matrix can be considered as acceptable [26].

*2.5. FCEM*

The term fuzzy logic was introduced by the Iranian-Azerbaijani mathematician Lotfi Zadeh in 1965 with his proposal of fuzzy set theory [27]. It is based on the observation that people make decisions based on imprecise and non-numerical information. Fuzzy models or sets are mathematical methods for representing fuzzy and imprecise information. These models can identify, represent, manipulate, interpret, and use data and information that is fuzzy and lacking in certainty [28].

FCEM is a comprehensive evaluation method based on fuzzy mathematics, which aims to convert fuzzy and qualitative evaluation into quantitative evaluation based on affiliation theory. It makes a comprehensive evaluation of the object that receives multiple factors constraints. In this study, because it is difficult for visitors to precisely define and describe the term "Smart", the FCEM method with fuzzy judgment is used [29].

This FCEM calculation process is performed using MATLAB. The process is divided into two steps. The first step is to establish the fuzzy judgment matrix.

The degree of membership of the item set Rm can be defined as:

$$Rm = \begin{bmatrix} R_{m1a} & R_{m1b} & \cdots & R_{m1e} \\ R_{m2a} & R_{m2b} & \cdots & R_{m2e} \\ \vdots & \vdots & \ddots & \vdots \\ R_{mna} & R_{mnb} & \cdots & R_{mne} \end{bmatrix}$$

The weighting of Item set A of the first classification calculated by AHP can be defined as:

$$A = \begin{bmatrix} A_1 & A_2 & A_3 & A_4 & A_5 \end{bmatrix}$$

The weighting of Item set $W_m$ of the secondary classification calculated by AHP can be defined as:

$$W_m = \begin{bmatrix} W_{m1} & W_{m2} & \cdots & W_{mn} \end{bmatrix}$$

where m represents the primary classification category, n represents the number of items in the secondary classification, and a–e corresponds to the 1–5 rating system. The matrix R corresponds to the raw data collected by the questionnaire.

The second step is to use the established matrix for the fuzzy comprehensive evaluation calculation as follows:

$$C_1 = W_1 \times R_1$$

$$C_2 = W_2 \times R_2$$

$$C_3 = W_3 \times R_3$$

$$C_4 = W_4 \times R_4$$

$$C_5 = W_5 \times R_5$$

$$B = A \times \begin{bmatrix} C_1 & C_2 & C_3 & C_4 & C_5 \end{bmatrix} = \begin{bmatrix} b_1 & b_2 & b_3 & b_4 & b_5 \end{bmatrix}$$

The bi value is the membership degree value of the evaluation item to each evaluation criterion and corresponds to the evaluation statement ("excellent", "good", "moderate", "fair", and "poor") according to the ranking. The maximum value is the result of this calculation.

*2.6. IPA*

IPA is a simple and practical satisfaction evaluation model that analyzes the gap between expectations and actual perceptions [30]. The four-quadrant diagram can help researchers quickly identify the key to the problem, distinguish the priority of each demand indicator, and thereby develop a targeted implementation plan.

The mean value of each item data of the original questionnaire was calculated, and the overall performance and importance mean values were taken as quadrant dividers. Finally, the position and stage of each item follows that shown in Figure 1.

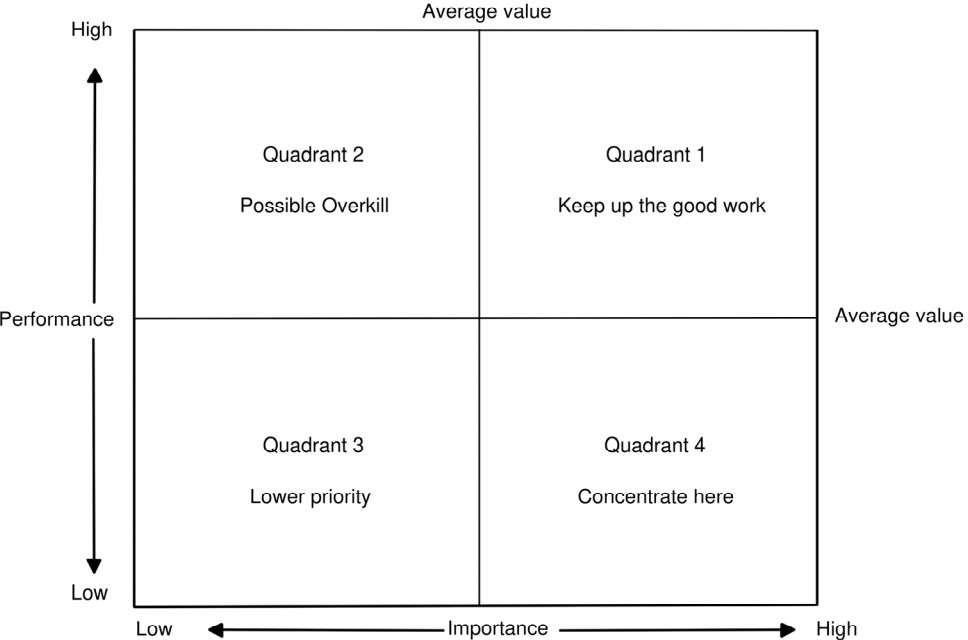

**Figure 1.** IPA matrix.

## 3. Results

*3.1. Results of AHP*

The importance ranking of the questionnaire and the defined factor are shown in Table 3.

The AHP method of pair-wise comparison of all items based on importance yields a judgment matrix of each factor. The judgment matrix of SZSOJ's first-level evaluation factors is shown in Table 4.

The judgment matrix of SZSOJ's second-level evaluation factors are shown in Tables 5–9.

All of the results passed the consistency test, indicating that the weight set obtained by AHP is reasonable.

The weight data obtained from AHP are generally in line with expectations, but there are still some differences. For example, the weight set of the Official website function at the first level of classification is 44.891, which is much higher than expected. The weights set of the Message notification (e-mail from Tokyo Zoonet) at the secondary level of classification is only 3.862, much lower than expected. Moreover, the weights set for

Stamps at the secondary level of classification is 6.746, which is not very high compared with the expected popular functions.

**Table 3.** The importance ranking of the questionnaire.

| Primary Classification | Result of Importance | Factor Defined | Rank | Secondary Classification | Result of Importance | Factor Defined | Rank |
|---|---|---|---|---|---|---|---|
| Official website function | 41.00% | $U_1$ | 1 | Electronic ticket system | 57.00% | $U_{17}$ | 1 |
| | | | | Online booking | 55.00% | $U_{16}$ | 2 |
| | | | | Event notice | 50.00% | $U_{13}$ | 3 |
| | | | | Official website function | 47.00% | $U_{11}$ | 4 |
| | | | | Online information access | 45.00% | $U_{12}$ | 5 |
| | | | | Tourism blog | 38.00% | $U_{18}$ | 6 |
| | | | | Electronic claim system | 23.00% | $U_{15}$ | 7 |
| | | | | Message notification (e-mail from Tokyo Zoonet) | 20.00% | $U_{14}$ | 8 |
| Tokyo Zoonet function | 12.00% | $U_2$ | 4 | Animal introduction platform | 39.00% | $U_{21}$ | 1 |
| | | | | 3D model | 27.00% | $U_{23}$ | 2 |
| | | | | VR sightseeing | 25.00% | $U_{22}$ | 3 |
| Smart (Tokyo Parks Navi) phone application(s) | 14.00% | $U_3$ | 3 | Digital map | 57.00% | $U_{32}$ | 1 |
| | | | | Obtain information from QR code | 43.00% | $U_{35}$ | 2 |
| | | | | Sightseeing route recommendation | 37.00% | $U_{33}$ | 3 |
| | | | | Animal information service | 37.00% | $U_{36}$ | 4 |
| | | | | Application | 34.00% | $U_{31}$ | 5 |
| | | | | Stamp | 26.00% | $U_{34}$ | 6 |
| Smartphone website application(s) (Tokyo Zoovie) | 5.00% | $U_4$ | 5 | Tourist spots recommendations | 40.00% | $U_{42}$ | 1 |
| | | | | Smart guide system | 37.00% | $U_{41}$ | 2 |
| Functions within the zoo | 26.00% | $U_5$ | 2 | Digital payment | 77.00% | $U_{52}$ | 1 |
| | | | | Free Wi-Fi | 60.00% | $U_{51}$ | 2 |
| | | | | Animal state observation (Monitoring) | 53.00% | $U_{55}$ | 3 |
| | | | | Electronic information screen | 43.00% | $U_{54}$ | 4 |
| | | | | Fundraising | 29.00% | $U_{53}$ | 5 |

**Table 4.** Judgment matrix of SZSOJ's first-level evaluation factors.

| Factor | $U_1$ | $U_2$ | $U_3$ | $U_4$ | $U_5$ | Weight (%) |
|---|---|---|---|---|---|---|
| $U_1$ | 1 | 4 | 4 | 5 | 2 | 44.891 |
| $U_2$ | 0.25 | 1 | 1 | 2 | 0.5 | 12.329 |
| $U_3$ | 0.25 | 1 | 1 | 2 | 0.5 | 12.329 |
| $U_4$ | 0.2 | 0.5 | 0.5 | 1 | 0.333 | 7.173 |
| $U_5$ | 0.5 | 2 | 2 | 3 | 1 | 23.279 |

$\lambda_{max}$ = 5.026, CI = 0.007, RI = 1.11, CR = 0.006 < 0.10.

**Table 5.** Judgment matrix of SZSOJ's second-level evaluation factors ($U_1$).

| Factor | $U_{11}$ | $U_{12}$ | $U_{13}$ | $U_{14}$ | $U_{15}$ | $U_{16}$ | $U_{17}$ | $U_{18}$ | Weight (%) |
|---|---|---|---|---|---|---|---|---|---|
| $U_{11}$ | 1 | 1 | 1 | 4 | 3 | 0.5 | 0.5 | 2 | 13.183 |
| $U_{12}$ | 1 | 1 | 1 | 4 | 3 | 0.5 | 0.5 | 2 | 13.183 |
| $U_{13}$ | 1 | 1 | 1 | 4 | 3 | 0.5 | 0.5 | 2 | 13.183 |
| $U_{14}$ | 0.25 | 0.25 | 0.25 | 1 | 1 | 0.25 | 0.25 | 0.333 | 3.862 |
| $U_{15}$ | 0.333 | 0.333 | 0.333 | 1 | 1 | 0.25 | 0.25 | 0.5 | 4.526 |
| $U_{16}$ | 2 | 2 | 2 | 4 | 4 | 1 | 1 | 3 | 22.17 |
| $U_{17}$ | 2 | 2 | 2 | 4 | 4 | 1 | 1 | 3 | 22.17 |
| $U_{18}$ | 0.5 | 0.5 | 0.5 | 3 | 2 | 0.333 | 0.333 | 1 | 7.724 |

$\lambda_{max}$ = 8.113, CI = 0.016, RI = 1.404, CR = 0.012 < 0.10.

**Table 6.** Judgment matrix of SZSOJ's second-level evaluation factors ($U_2$).

| Factor | $U_{21}$ | $U_{22}$ | $U_{23}$ | Weight (%) |
|---|---|---|---|---|
| $U_{21}$ | 1 | 2 | 2 | 50 |
| $U_{22}$ | 0.5 | 1 | 1 | 25 |
| $U_{23}$ | 0.5 | 1 | 1 | 25 |

$\lambda_{max}$ = 3.0, CI = 0.0, RI = 0.525, CR = 0.0 < 0.10.

**Table 7.** Judgment matrix of SZSOJ's second-level evaluation factors ($U_3$).

| Factor | $U_{31}$ | $U_{32}$ | $U_{33}$ | $U_{34}$ | $U_{35}$ | $U_{36}$ | Weight (%) |
|---|---|---|---|---|---|---|---|
| $U_{31}$ | 1 | 0.333 | 1 | 2 | 0.5 | 1 | 12.019 |
| $U_{32}$ | 3 | 1 | 3 | 4 | 2 | 3 | 35.357 |
| $U_{33}$ | 1 | 0.333 | 1 | 2 | 0.5 | 1 | 12.019 |
| $U_{34}$ | 0.5 | 0.25 | 0.5 | 1 | 0.333 | 0.5 | 6.746 |
| $U_{35}$ | 2 | 0.5 | 2 | 3 | 1 | 2 | 21.84 |
| $U_{36}$ | 1 | 0.333 | 1 | 2 | 0.5 | 1 | 12.019 |

$\lambda_{max}$ = 6.032, CI = 0.006, RI = 1.25, CR = 0.005 < 0.10.

**Table 8.** Judgment matrix of SZSOJ's second-level evaluation factors ($U_4$).

| Factor | $U_{41}$ | $U_{42}$ | Weight (%) |
|---|---|---|---|
| $U_{41}$ | 1 | 1 | 50 |
| $U_{42}$ | 1 | 1 | 50 |

$\lambda_{max}$ = 2, CI = 0.0, RI = 0, CR = 0.0 < 0.10.

**Table 9.** Judgment matrix of SZSOJ's second-level evaluation factors ($U_5$).

| Factor | $U_{51}$ | $U_{52}$ | $U_{53}$ | $U_{54}$ | $U_{55}$ | Weight (%) |
|---|---|---|---|---|---|---|
| $U_{51}$ | 1 | 0.333 | 4 | 3 | 2 | 24.34 |
| $U_{52}$ | 3 | 1 | 4 | 4 | 3 | 43.388 |
| $U_{53}$ | 0.25 | 0.25 | 1 | 0.5 | 0.333 | 6.445 |
| $U_{54}$ | 0.333 | 0.25 | 2 | 1 | 0.5 | 9.769 |
| $U_{55}$ | 0.5 | 0.333 | 3 | 2 | 1 | 16.058 |

$\lambda_{max}$ = 5.170, CI = 0.043, RI = 1.11, CR = 0.038 < 0.10.

### 3.2. Results of FCEM

The specific values for each second-level evaluation factor of the questionnaire are shown in Tables 10–14.

**Table 10.** Results (Performance) of the second-level questionnaire (Official website function).

| Factor | Score | | | | |
|---|---|---|---|---|---|
| | 1 | 2 | 3 | 4 | 5 |
| Official website function | 50.50% | 12.90% | 9.90% | 13.90% | 12.90% |
| Online information access | 27.70% | 20.80% | 14.90% | 15.80% | 20.80% |
| Event notice | 31.70% | 18.80% | 21.80% | 11.90% | 15.80% |
| Message notification (e-mail from Tokyo Zoonet) | 41.60% | 19.80% | 15.80% | 10.90% | 11.90% |
| Electronic claim system | 50.50% | 14.90% | 13.90% | 7.90% | 12.90% |
| Electronic ticket system | 30.70% | 12.90% | 17.80% | 11.90% | 26.70% |
| Online booking | 28.70% | 9.90% | 13.90% | 17.80% | 29.70% |
| Tourism blog | 27.70% | 19.80% | 23.80% | 12.90% | 15.80% |

**Table 11.** Results (Performance) of the second-level questionnaire (Tokyo Zoonet function).

| Factor | Score | | | | |
|---|---|---|---|---|---|
| | 1 | 2 | 3 | 4 | 5 |
| Animal introduction platform | 32.70% | 13.90% | 25.70% | 15.80% | 11.90% |
| VR sightseeing | 48.50% | 12.90% | 19.80% | 9.90% | 8.90% |
| 3D model | 46.50% | 13.90% | 15.80% | 12.90% | 10.90% |

**Table 12.** Results (Performance) of the second-level questionnaire (Smartphone application(s)).

| Factor | Score | | | | |
|---|---|---|---|---|---|
| | 1 | 2 | 3 | 4 | 5 |
| Application | 50.50% | 9.90% | 13.90% | 11.90% | 13.90% |
| Digital map | 24.80% | 11.90% | 18.80% | 15.80% | 28.70% |
| Sightseeing route recommendation | 32.70% | 17.80% | 18.80% | 14.90% | 15.80% |
| Stamp | 43.60% | 12.90% | 16.80% | 11.90% | 14.90% |
| Information from QR code | 31.70% | 15.80% | 14.90% | 12.90% | 24.80% |
| Animal information service | 35.60% | 12.90% | 16.80% | 15.80% | 18.80% |

**Table 13.** Results (Performance) of the second-level questionnaire (Smartphone website application(s)).

| Factor | Score | | | | |
|---|---|---|---|---|---|
| | 1 | 2 | 3 | 4 | 5 |
| Smart guide system | 40.60% | 11.90% | 13.90% | 10.90% | 22.80% |
| Tourist spots recommendations | 38.60% | 8.90% | 18.80% | 13.90% | 19.80% |

**Table 14.** Results (Performance) of the second-level questionnaire (Functions within the zoo).

| Factor | Score | | | | |
|---|---|---|---|---|---|
| | 1 | 2 | 3 | 4 | 5 |
| Free Wi-Fi | 26.70% | 13.90% | 15.80% | 10.90% | 32.70% |
| Digital payment | 12.90% | 9.90% | 7.90% | 13.90% | 55.40% |
| Fundraising | 42.60% | 16.80% | 16.80% | 13.90% | 9.90% |
| Electronic information screen | 19.80% | 20.80% | 21.80% | 10.90% | 26.70% |
| Animal state observation (Monitoring) | 20.80% | 10.90% | 20.80% | 17.80% | 29.70% |

The weighting of Item set A of the first classification and the weighting of Item set $W_m$ of the secondary classification calculated by AHP are shown as follows:

$A = \begin{bmatrix} 0.4489 & 0.1233 & 0.1233 & 0.0717 & 0.232 \end{bmatrix}$

$W_1 = \begin{bmatrix} 0.1318 & 0.1318 & 0.1318 & 0.0386 & 0.0453 & 0.2217 & 0.2217 & 0.0772 \end{bmatrix}$

$W_2 = \begin{bmatrix} 0.5000 & 0.2500 & 0.2500 \end{bmatrix}$

$W_3 = \begin{bmatrix} 0.1202 & 0.3536 & 0.1202 & 0.0675 & 0.2184 & 0.1202 \end{bmatrix}$

$W_4 = \begin{bmatrix} 0.5000 & 0.5000 \end{bmatrix}$

$W_5 = \begin{bmatrix} 0.2434 & 0.4339 & 0.0645 & 0.0977 & 0.1606 \end{bmatrix}$

According to the results (Performance) of the second-level questionnaire, Item set $R_m$ can be constructed as follows:

$$R_1 = \begin{bmatrix} 0.13 & 0.14 & 0.10 & 0.13 & 0.51 \\ 0.21 & 0.16 & 0.15 & 0.21 & 0.28 \\ 0.16 & 0.12 & 0.22 & 0.19 & 0.32 \\ 0.12 & 0.11 & 0.16 & 0.20 & 0.42 \\ 0.13 & 0.08 & 0.14 & 0.15 & 0.51 \\ 0.27 & 0.12 & 0.18 & 0.13 & 0.31 \\ 0.30 & 0.18 & 0.14 & 0.10 & 0.29 \\ 0.16 & 0.13 & 0.24 & 0.20 & 0.28 \end{bmatrix}$$

$$R_2 = \begin{bmatrix} 0.12 & 0.16 & 0.26 & 0.14 & 0.33 \\ 0.09 & 0.10 & 0.20 & 0.13 & 0.49 \\ 0.11 & 0.13 & 0.16 & 0.14 & 0.47 \end{bmatrix}$$

$$R_3 = \begin{bmatrix} 0.14 & 0.12 & 0.14 & 0.10 & 0.51 \\ 0.29 & 0.16 & 0.19 & 0.12 & 0.25 \\ 0.16 & 0.15 & 0.19 & 0.18 & 0.33 \\ 0.15 & 0.12 & 0.17 & 0.13 & 0.44 \\ 0.25 & 0.13 & 0.15 & 0.16 & 0.32 \\ 0.19 & 0.16 & 0.17 & 0.13 & 0.36 \end{bmatrix}$$

$$R_4 = \begin{bmatrix} 0.23 & 0.11 & 0.14 & 0.12 & 0.41 \\ 0.20 & 0.14 & 0.19 & 0.09 & 0.39 \end{bmatrix}$$

$$R_5 = \begin{bmatrix} 0.33 & 0.11 & 0.16 & 0.14 & 0.27 \\ 0.55 & 0.14 & 0.08 & 0.10 & 0.13 \\ 0.10 & 0.14 & 0.17 & 0.17 & 0.43 \\ 0.27 & 0.11 & 0.22 & 0.21 & 0.20 \\ 0.30 & 0.18 & 0.21 & 0.11 & 0.21 \end{bmatrix}$$

By using assessment matrix C and the corresponding weight vector A, the result of the first-level fuzzy comprehensive evaluation can be obtained by using $B = A \times C$.

$$B = \begin{bmatrix} 0.2466 & 0.1382 & 0.1663 & 0.1384 & 0.3196 \end{bmatrix}$$

The results of fuzzy comprehensive evaluation are usually defined according to the maximum membership degree principle. From vector B, it can be recognized that the membership degree values of "excellent", "good", "moderate", "fair", and "poor" are 0.2466, 0.1382, 0.1663, 0.1384, and 0.3196, respectively. Among them, the membership degree value of "poor" (0.3169) is the largest. Thus, the SZSOJ evaluation score of Ueno Zoo is 0.288 (at the level of "fair"). This indicates that the intellectualization construction in Ueno Zoo is still in the initial stage, and the level of intellectualization is relatively low. This means that the visitors do not effectively feel the impact of "Smart".

*3.3. Results of IPA*

The mean values of all factors can be calculated by SPSS from the raw data of the questionnaire, and are shown in Tables 15 and 16. The resulting computed IPA matrices are shown in Figures 2 and 3.

**Table 15.** Means and ranking of items at the first level of classification.

| Factor | Performance (P) | | Importance (I) | | Mean Difference (P-I) | t Value | p Value |
|---|---|---|---|---|---|---|---|
| | Average Value | Rank | Average Value | Rank | | | |
| Official website function | 2.61 | 4 | 3.84 | 3 | −1.23 | −11.142 | <0.001 |
| Tokyo Zoonet function | 2.35 | 5 | 3.54 | 5 | −1.18 | −10.689 | <0.001 |
| Smart (Tokyo Parks Navi) phone application(s) | 2.66 | 2 | 3.82 | 4 | −1.16 | −10.02 | <0.001 |
| Smartphone website application(s) (Tokyo Zoovie) | 2.65 | 3 | 3.87 | 2 | −1.21 | −8.243 | <0.001 |
| Functions within the zoo | 3.12 | 1 | 4.10 | 1 | −0.98 | −10.005 | <0.001 |

**Table 16.** Means and ranking of items at the second level of classification.

| Factor | Performance (P) | | Importance (I) | | Mean Difference (P-I) | t Value | p Value |
|---|---|---|---|---|---|---|---|
| | Average Value | Rank | Average Value | Rank | | | |
| Official website function | 2.26 | 22 | 4 | 8 | −1.743 | −11.091 | <0.001 |
| Online information access | 2.81 | 9 | 3.9 | 13 | −1.089 | −7.152 | <0.001 |
| Event notice | 2.61 | 15 | 4.11 | 7 | −1.495 | −9.967 | <0.001 |
| Message notification (e-mail from Tokyo Zoonet) | 2.32 | 18 | 3.25 | 23 | −0.931 | −6.886 | <0.001 |
| Electronic claim system | 2.18 | 23 | 3.38 | 22 | −1.198 | −7.495 | <0.001 |
| Electronic ticket system | 2.91 | 7 | 4.2 | 4 | −1.287 | −8.444 | <0.001 |
| Online booking | 3.1 | 4 | 4.18 | 5 | −1.079 | −7.04 | <0.001 |
| Tourism blog | 2.69 | 10 | 3.7 | 17 | −1.01 | −8.049 | <0.001 |
| Animal introduction platform | 2.6 | 16 | 3.78 | 16 | −1.178 | −8.758 | <0.001 |
| VR sightseeing | 2.18 | 24 | 3.42 | 20 | −1.238 | −9.368 | <0.001 |
| 3D model | 2.28 | 21 | 3.42 | 21 | −1.139 | −8.481 | <0.001 |
| Application | 2.29 | 20 | 3.57 | 18 | −1.287 | −8.444 | <0.001 |
| Digital map | 3.12 | 3 | 4.27 | 2 | −1.149 | −7.631 | <0.001 |
| Sightseeing route recommendation | 2.63 | 13 | 3.94 | 12 | −1.307 | −9.032 | <0.001 |
| Stamps | 2.42 | 17 | 3.21 | 24 | −0.792 | −5.858 | <0.001 |
| Information from QR code | 2.83 | 8 | 3.98 | 9 | −1.149 | −7.699 | <0.001 |
| Animal information service | 2.69 | 11 | 3.95 | 11 | −1.257 | −8.907 | <0.001 |
| Smart guide system | 2.63 | 14 | 3.85 | 15 | −1.218 | −7.631 | <0.001 |
| Tourist spots recommendations | 2.67 | 12 | 3.88 | 14 | −1.208 | −7.925 | <0.001 |
| Free Wi-Fi | 3.09 | 5 | 4.23 | 3 | −1.139 | −7.544 | <0.001 |
| Digital payment | 3.89 | 1 | 4.61 | 1 | −0.723 | −5.441 | <0.001 |
| Fundraising | 2.32 | 19 | 3.52 | 19 | −1.208 | −8.613 | <0.001 |
| Electronic information screen | 3.04 | 6 | 3.98 | 10 | −0.941 | −7.626 | <0.001 |
| Animal state observation (Monitoring) | 3.25 | 2 | 4.16 | 6 | −0.911 | −6.782 | <0.001 |

As seen in Figure 2, functions within the zoo in the first quadrant are the most recognized and valued by visitors and should be continually maintained. The Tokyo Zoonet function (in the third quadrant), which is not valued and has low recognition, can be placed in a lower priority development plan, whereas Smartphone applications (featured in the third quadrant) is in a slightly better situation, and continued investment is expected to steer this item towards the other quadrants. The remaining two second-quadrant features feature limited construction and require further investment due to demands received from visitors.

As seen in Figure 3, Digital payment in the first quadrant is the most recognized and valued by visitors, and some other functions such as Animal state observation, Digital map, Electronic information screen, Free Wi-Fi, Online booking, and Electronic ticket system need to be continually maintained. Functions in the third quadrant include Stamps, Message notification, Electronic claim system, VR sightseeing, 3D models, Application, and Fundraising, which are not valued and have low recognition; these can be placed in a lower priority development plan. Functions in the fourth quadrant, such as the Official website function and Event notice, have limited construction and require further investment due to demand from visitors. The remaining features gathered around the center of four quadrants are yet to be promoted and lack investment for enhancing the smart experience of visitors.

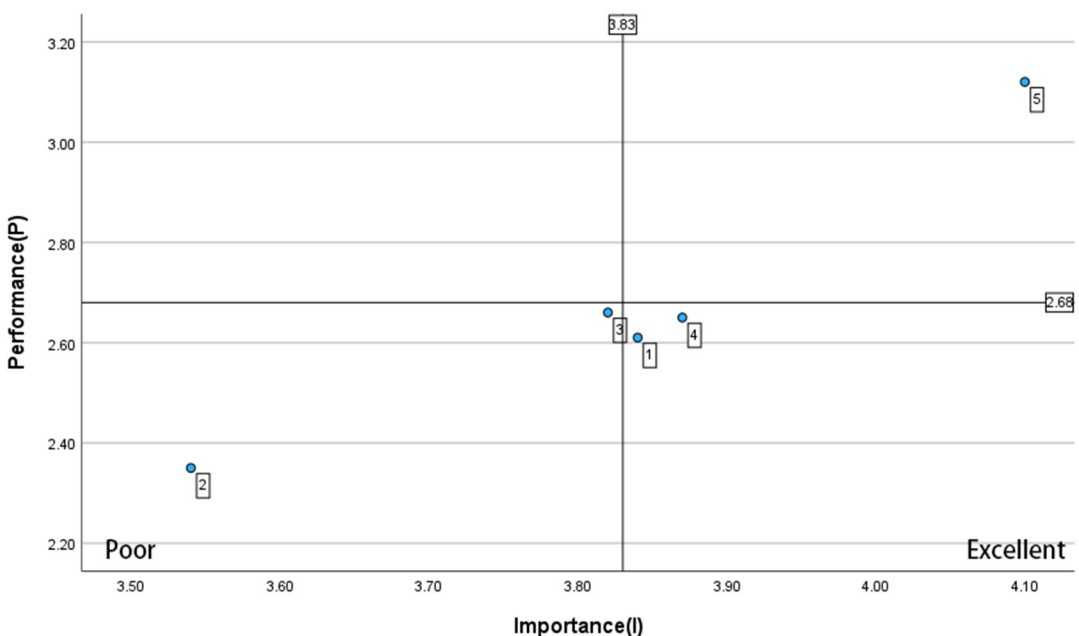

**Figure 2.** IPA matrix of items at the first level of classification. Note: 1. Official website function; 2. Tokyo Zoonet function; 3. Smart (Tokyo Parks Navi) phone application(s); 4. Smart phone website ap-plication(s) (Tokyo Zoovie); 5. Functions within the zoo.

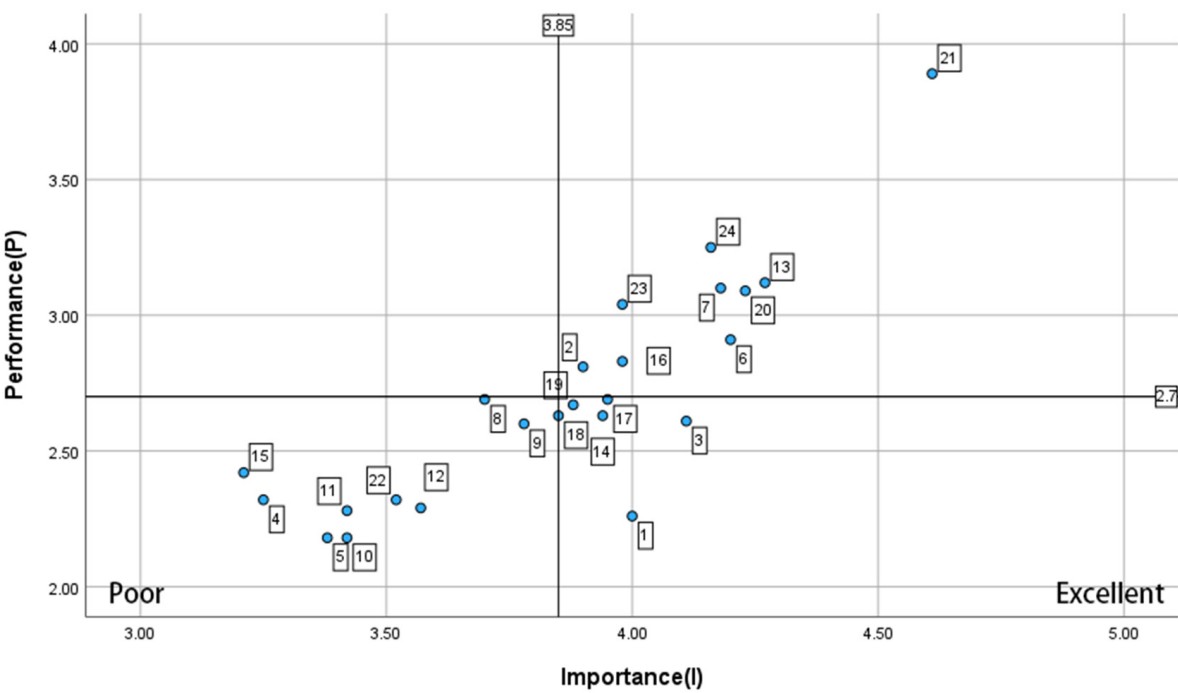

**Figure 3.** IPA matrix of items at the second level of classification. Note: 1. Official website function; 2. Online information access; 3. Event notice; 4. Message notification (e-mail from Tokyo Zoonet); 5. Electronic claim system; 6. Electronic ticket system; 7. Online booking; 8. Tourism blog; 9. Animal introduction platform; 10. VR sightseeing; 11. 3D models; 12. Application; 13. Digital map; 14. Sightseeing route recommendation; 15. Stamp; 16. Obtaining information from QR code; 17. Animal information service; 18. Smart guide system; 19. Tourist spots recommendations; 20. Free Wi-Fi; 21. Digital payment; 22. Fundraising; 23. Electronic information screen; 24. Animal state observation (Moni-toring).

## 4. Discussion

### 4.1. Findings from Questionnaire

The results of the questionnaire in our study showed that the overall performance aspect of Ueno Zoo was not very good, with the mean scores for many items being lower than expected, whereas the scores for the importance aspect were better than expected and produced greater variability. The reasons for this phenomenon may depend on the following reasons:

1.  The smart functions of Ueno Zoo are not well publicized [31], and it is difficult for ordinary visitors to access them apart from some essential functions such as online booking and the electronic ticket system. In the process of collecting questionnaires, a common theme of the feedback received was "The existence of these functions is unknown without the questionnaire graphics", so the feedback on the zoo's performance based on the questionnaire was not good.
2.  The zoo itself is a special type of park that places more emphasis on offline interaction, so the best feedback would come from features where one can interact with the venue [32]. At the same time, some zoos and schools that promote educational knowledge, such as animal information services, will also receive attention from zoos and citizens, and therefore, will also receive good feedback. However, some functions related to overall zoo management and construction, such as 3D models, are not important to visitors, and thus, do not receive good feedback.
3.  Concerns exist within Japan, especially Tokyo—due to the COVID-19 pandemic [33], declining birthrate, and aging population—that the smart-city process may eliminate or reduce drawbacks associated with traditional methods of production [34]. The overall feedback on the importance aspect of the questionnaire was good; this may be because the impact of intellectualization on ordinary citizens can make people realize the importance of intellectualization.

### 4.2. Findings from Analytical Calculdations

The results of FCEM show that Ueno Zoo is still in the initial stage (the FCEM evaluation score of Ueno Zoo is 0.3196 at the level of "fair") of intellectualization construction; the level of intellectualization is relatively low and visitors are not well informed about it. However, this result is not too bad. On the one hand, this process has not been implemented for a long period and it requires people to slowly adapt to it. On the other hand, due to the COVID-19-related visitor-traffic decline, people still need some time to become familiarized with the various "Smart" changes in the zoo. The zoo needs more publicity and a more "Smart" and portable way to integrate intelligent tourism into daily life.

The weight set of the evaluation items calculated by AHP in this study reveals the order in which SZSOJ was constructed. The development of zoos requires a great deal of effort, time, and resources. Evaluating the weight set of the projects allows managers to understand the roles that these key projects play in the development of SZSOJ.

Here we have created a SWOT chart to summarize the view in Figure 4.

Based on the current state, we think the current state of smart zoo construction in Japan is still in the initial stage, so it is a challenge full of opportunities for Ueno Zoo. If the managers take advantage of the zoo's regional traffic and existing construction level, it will be a pioneering case that will greatly contribute to the smart construction of the region and the ability of other zoos in Japan to alleviate current social problems.

Based on this conclusion, we provide our opinion on the change in the framework for the development of the intellectualization of the Ueno Zoo in Figures 5 and 6.

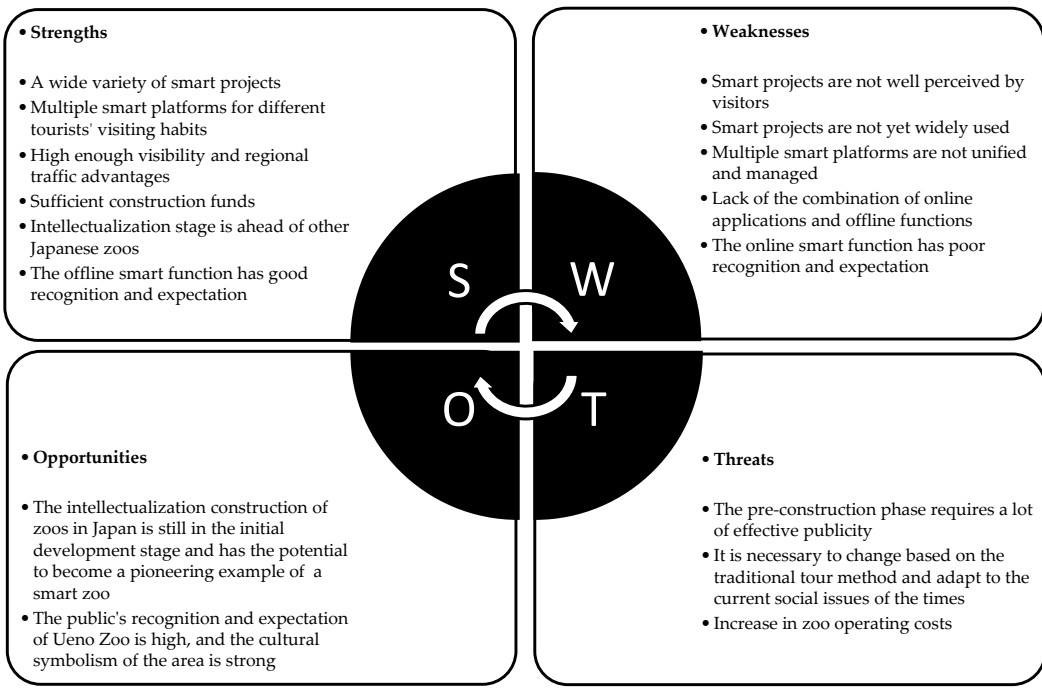

**Figure 4.** SWOT of the current state of Ueno Zoo.

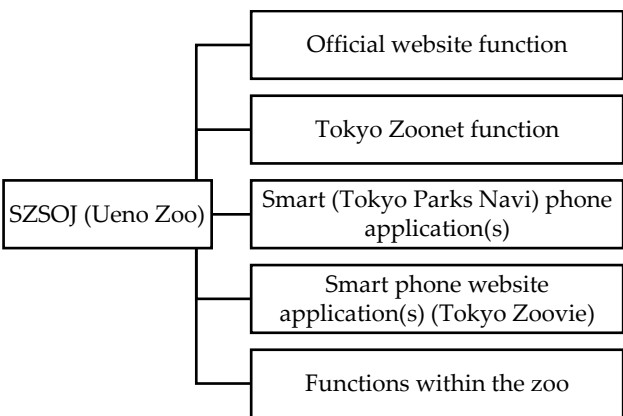

**Figure 5.** The framework system of the current state of Ueno Zoo.

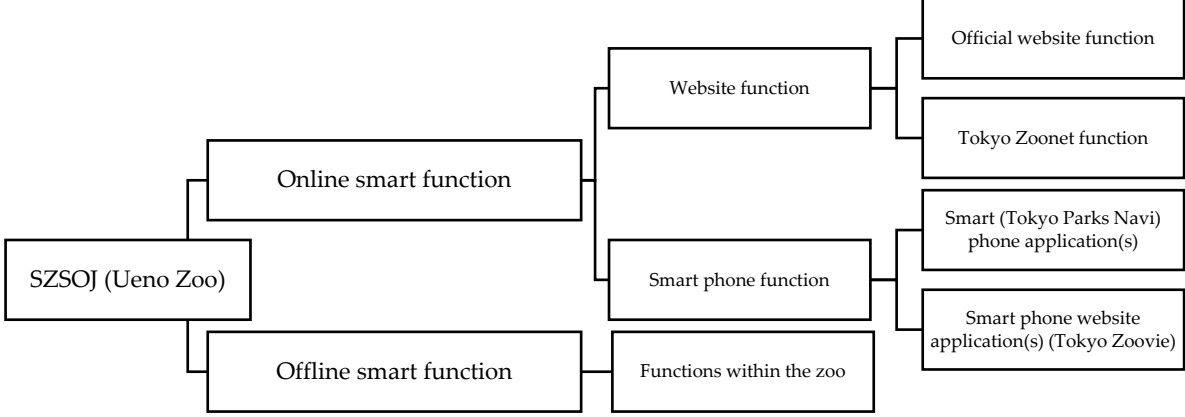

**Figure 6.** The suggested framework system of the future state of Ueno Zoo.

## 5. Conclusions

The main purpose of this study is to determine the actual degree of intellectualization construction in Japanese zoos through the analysis method of FCEM, in which the weights are determined through AHP, as well as the current strengths and weaknesses of zoos' intellectualization function development and development prospects through IPA. Smart zoos are an emerging topic, especially in Japan, which is in the midst of smart-city development. This study aims to establish a method to more objectively judge the intellectualization process with ambiguous expressions, and to help zoos in Japan—and even the world—become Smarter. Our results could be utilized for comparisons with current policies as well as guiding future development.

However, this study still has some limitations, which will be useful for guiding future research. First, the selection of this smart project was changed due to some characteristics of Ueno Zoo. For example, the Ueno Zoo site is in the territory of Ueno Park and there is no parking lot, so some features (such as intelligent transportation) were not taken into consideration. Secondly, even though the popularity of smartphones has boomed in the last 10 years, many elderly people are not familiar with this system, and this has an impact on the perception of wisdom or 'smart'. This phenomenon also shows the importance of a smart way to visit a city in the future without the hindrance of devices. Future research will also consider more smart features within the venue. In addition, only Ueno Zoo, one of the four members of the Tokyo Zoological Park Society, was studied. It is possible that a comparison with other zoos under a unified management system and the use of AHP to make decisions would enrich the breadth of the study.

**Supplementary Materials:** The following supporting information can be downloaded at: https://www.mdpi.com/article/10.3390/land12010243/s1, Supplementary File S1: The following is the supplementary data related to this article.

**Author Contributions:** Conceptualization, Y.L. and R.S.; methodology, Y.L. and R.S.; software, Y.L.; validation, Y.L.; formal analysis, Y.L.; investigation, Y.L.; resources, Y.L.; data curation, Y.L.; writing—original draft preparation, Y.L.; writing—review and editing, Y.L. and R.S.; visualization, Y.L.; supervision, Y.L. and R.S.; project administration, Y.L.; funding acquisition, Y.L. and R.S. All authors have read and agreed to the published version of the manuscript.

**Funding:** This research received no external funding.

**Data Availability Statement:** Not applicable.

**Conflicts of Interest:** The authors declare no conflict of interest.

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
