# Peer review of "Impact of Intellectualization of a Zoo through a FCEM-AHP and IPA Approach"

_land, doi:10.3390/land12010243_

Round 1
Reviewer 1 Report
Dear authors,
Research item is interesting. The methodological part of the research is structured well.
It will be fine to update research in the analysis of the previous researchs with the research papers of the newer date.
Author Response
Dear Reviewer,
Thank you for your pay attention to this paper.
We have added and revised a new section in Part I (Introduction) and revised the issues raised to focus on getting feedback on the smart zoo building process timely and exploring a stable way to understand the current level of zoo Intellectualization and improvement measures.
And in the conclusion section, we added the opinion in the form of SWOT and the framework idea for the smart zoo to better build the smart process of Ueno Zoo.

Reviewer 2 Report
This paper represents a very good scientific methodology. Useful for others researches.
Author Response
Thank you for taking the time to consider our article. We thank you for your helpful comments and constructive suggestions.

Reviewer 3 Report
The article submitted for review refers to very interesting issues related to the functioning of smart parks systems. Therefore, the topic is current and important for the modern functioning of zoological parks. In the first part, the authors presented a research problem that, in my opinion, is not sufficiently described and should be corrected in such a way as to be compatible with the presented research. However, the research carried out is reliable and can actually significantly support the management system of this type of land. On the basis of research and e.g. SWOT analysis, it would be possible to present the main structural elements of the intelligent database synchronization system with the expectations of tourists, and thus determine the framework system for designing a smart system in this zoo. It would also be good to refer to the differences between smart parks and smart zoological parks. This would be the added value of the conducted research.
I propose to redraft the first part of the text as well as the discussion and conclusions.
Author Response

(The authors gave the same response as above.)
